# Credentialing and Pharmacologically Targeting PTP4A3 Phosphatase as a Molecular Target for Ovarian Cancer

**DOI:** 10.3390/biom11070969

**Published:** 2021-06-30

**Authors:** John S. Lazo, Elizabeth R. Sharlow, Robert Cornelison, Duncan J. Hart, Danielle C. Llaneza, Anna J. Mendelson, Ettore J. Rastelli, Nikhil R. Tasker, Charles N. Landen, Peter Wipf

**Affiliations:** 1Department of Pharmacology, University of Virginia, Charlottesville, VA 22908, USA; jrc3hg@virginia.edu (R.C.); djh2kd@virginia.edu (D.J.H.); dcl5z@virginia.edu (D.C.L.); ajm2ty@virginia.edu (A.J.M.); 2KeViRx, Inc., Charlottesville, VA 22904, USA; 3Department of Chemistry, University of Pittsburgh, Pittsburgh, PA 15260, USA; erastelli@ptcbio.com (E.J.R.); NRT22@pitt.edu (N.R.T.); pwipf@pitt.edu (P.W.); 4Department of Obstetrics and Gynecology, University of Virginia, Charlottesville, VA 22908, USA; cl3nj@virginia.edu

**Keywords:** ovarian cancer, protein tyrosine phosphatase, drug target validation, cell migration, IL-6, drug synergy

## Abstract

High grade serous ovarian cancer (OvCa) frequently becomes drug resistant and often recurs. Consequently, new drug targets and therapies are needed. Bioinformatics-based studies uncovered a relationship between high Protein Tyrosine Phosphatase of Regenerating Liver-3 (PRL3 also known as PTP4A3) expression and poor patient survival in both early and late stage OvCa. PTP4A3 mRNA levels were 5–20 fold higher in drug resistant or high grade serous OvCa cell lines compared to nonmalignant cells. JMS-053 is a potent allosteric small molecule PTP4A3 inhibitor and to explore further the role of PTP4A3 in OvCa, we synthesized and interrogated a series of JMS-053-based analogs in OvCa cell line-based phenotypic assays. While the JMS-053 analogs inhibit in vitro PTP4A3 enzyme activity, none were superior to JMS-053 in reducing high grade serous OvCa cell survival. Because PTP4A3 controls cell migration, we interrogated the effect of JMS-053 on this cancer-relevant process. Both JMS-053 and CRISPR/Cas9 PTP4A3 depletion blocked cell migration. The inhibition caused by JMS-053 required the presence of PTP4A3. JMS-053 caused additive or synergistic in vitro cytotoxicity when combined with paclitaxel and reduced in vivo OvCa dissemination. These results indicate the importance of PTP4A3 in OvCa and support further investigations of the lead inhibitor, JMS-053.

## 1. Introduction

Annually, ~22,000 new cases of ovarian cancer (OvCa) are diagnosed in the US with the majority being late stage with a very poor predicted overall survival. While initial responses to surgical debulking and first line chemotherapy are high, drug-resistant tumor recurrence and dissemination are common leading to death. Thus, there is a significant need to identify mechanistically new molecular targets and drugs especially for high grade serous OvCa (HGSOC).

As with other cancers, OvCa is a disease driven by the dysregulation of intracellular signaling networks and we now understand many of the signaling processes by which cells transmit external information to internal effectors. Protein tyrosine kinases, which frequently become drivers of neoplastic disorders, play a prominent role. Covalent protein phosphorylation, however, is reversible and, thus, protein phosphatases function with kinases as non-redundant co-regulators of the overall protein phosphorylation status. Of the 125 human Tyr phosphatases (PTPs), >20 have been reported to participate in the control of aberrant pathways associated with cancer and in resistance to both conventional chemotherapy and targeted therapies [1]. Unlike Tyr kinases, which have been extensively interrogated and drugged, the PTPs have been more challenging therapeutic targets, in part, because their protein structures often lack obvious small molecule binding sites and they have important roles in normal tissues as well as tumors [2,3]. Recently, however, allosteric small molecule inhibitors of two PTPs, PTP1B and SHP2 phosphatase, have made significant advancements in clinical trials and this has generated considerable enthusiasm for PTPs as relevant and innovative molecular targets for antineoplastic interventions [4,5,6].

A major obstacle in understanding HGSOC transformation and progression is the limited number of common genetic alterations generally seen in patient populations. There are few common mutations outside of the initiating alteration to TP53, and the tumor is generally described as being a disease dominated by copy number alterations. The most common copy number change seen is amplification of 8q24 [7]. The MYC proto-oncogene is thought to be the relevant target of the amplification, but recent research is beginning to question that assumption [8,9,10]. There are other oncogenic genes located on 8q24, and one of these, Protein Tyrosine Phosphatase 4A3 (PTP4A3 also known as PRL-3), has gained some notoriety as being the most oncogenic phosphatase seen in human cancers [11]. Elevated levels of PTP4A3 are known to promote cellular invasion, motility, angiogenesis, and survival [12,13,14], which are properties commonly associated with highly malignant and disseminated cancers. PTP4A3 also alters cancer cell adhesion [15]. We previously reported that PTP4A3 was overexpressed in human OvCa tumor samples [16] but the involvement of PTP4A3 in HGSOC is less well understood. 

Efforts to uncover the biochemical functions of PTP4A3 and credential it as a high-value tumor target have been hampered due to the dearth of potent and selective inhibitors of PTP4A3. We previously described a scalable synthesis of the most potent known allosteric small molecule PTP4A3 inhibitor, 7-imino-2-phenylthieno [3,2-*c*]pyridine-4,6(5*H*,7*H*)-dione or JMS-053 [17,18] and benchmarked its cytotoxicity against eight chemosensitive and chemoresistant human OvCa cell lines grown as 3-dimensional spheroids including the taxane resistant SKOV3-TRip2 cells [16]. Importantly, no significant cell death was observed when normal fibroblast (IMR-90) and ovarian surface epithelial cells (HIO-180) were exposed to 25 µM JMS-053 [16].

In the current study, we uncovered a strong correlation between high PT4A3 expression and poor patient survival with HGSOC. Unexpectedly, we also determined that patients with early stage (i.e., I/II) HGSOC and high PTP4A3 expression also have a survival disadvantage, which is a novel finding. Moreover, we report on the synthesis of several new analogs of JMS-053 to explore further potential structural modifications of this core chemotype and provide information concerning the biochemical and cellular activity of JMS-053 and its analogs in HGSOC cells. Additionally, we document the ability of JMS-053 to reduce the dissemination of human drug-resistant OvCa in mice.

## 2. Materials and Methods

### 2.1. Cells and Reagents

OVCAR4 cells were purchased from Charles River (Frederick, MD, USA). OVCAR4-PTP4A3 CRISPR knock out cells were generated by Synthego (Menlo Park, CA, USA). A2780 and IMR-90 cells were purchased from the American Type Culture Collection [19] (Manassas, VA, USA). Kuramochi cells were purchased from Sekisui XenoTech, LLC (Kansas City, KS, USA). We previously described the SV40-immortalized nontumorigenic cell line HIO-180, which was derived from normal ovarian surface epithelium, HeyA8, HeyA8-MDR, SKOV3, taxane-resistant SKOVTRip2, and A2780cp20 cells [19], which were a gift from Anil Sood (M.D. Anderson Cancer Center, Houston, TX, USA). COV362, OV90, and OVSAHO cells were provided by Kenneth R. Nephew (Indiana University, Bloomington, IN, USA). COV362-47R cells were created over a one year period by exposing COV362 to increasing concentrations of carboplatin until a stable resistant line was created. The EC_50_ (concentration that caused a 50% cytotoxicity) for the COV362-47R cells using a viability assay in the presence of 3-(4,5-dimethylthiazol-2-yl)-2,5-diphenyltetrazolium bromide was ~200 μM for carboplatin compared to ~20 μM for the COV362 parent line. The V581 cell line was created by culturing tumor cells from the ascites fluid of a patient with recurrent and taxane-resistant OvCa. All cell lines were cultured in RPMI medium without antibiotics supplemented with 10% FBS with the exception of V581 cells, which were grown Ham’s F-12 medium supplemented with 20% FBS, 80 μM GlutaMAX, 20 μM sodium pyruvate, 0.02% B27, and 0.1 mg/mL Normocin. All cells were passaged < 22 times and then discarded. Cells were monitored at least every six months for *mycoplasma* contamination. The STAT3 pathway inhibitor 864,669 was synthesized as previously described [20]. Paclitaxel (catalog # HY-B0015) and olaparib (catalog # S1060) were from VWR (Radnor, PA, USA) and SHP099 was from Fisher Scientific (Waltham, MA, USA) (catalog # HY-100388A). All other reagents were purchased from Invitrogen (Waltham, MA, USA) unless otherwise noted. PTP4A3 mRNA expression in the Broad Cancer Cell Library Encyclopedia was obtained from data accessed on 31 June 2021 at https://portals.broadinstitute.org/ccle.

### 2.2. Clinical Gene Expression

KMplot (https://kmplot.com/analysis/) accessed on 31 June 2021 was used to generate OvCa survival plots, limiting the patient population only to serous OvCa to avoid the endometrial cancer subset. We used the median cutoff value for false discovery rate (FDR) and focused on overall survival (OS) as the primary metric. Beeswarm plots were generated from the gene expression data in the patient set used for the survival analysis. Median survival was calculated in cases where both of the patient groups reached a median survival, whereas in other cases upper quartile was used. Patients exceeding the threshold of survival were censored as opposed to being excluded from the analysis. The patient datasets were from the 1287 OvCa patients taken from both The Cancer Genome Atlas (TCGA) and The Gene Expression Omnibus (GEO) microarray data available online (http://www.cbioportal.org) accessed on 10 June 2021.

### 2.3. Synthesis and Chemical Characterization of Compounds

The synthesis and chemical characterizations of JMS-053 and NRT-870-59 have been previously described [17,18]. EJR-912-41, EJR-912-57, EJR-912-43, EJR-912-50 and NRT-902-38 were obtained analogously; experimental protocols and spectral information are listed are listed in the supplementary file.

### 2.4. In Vitro Phosphatase Inhibition

The cDNA for the full-length ptp4a3 was obtained from OriGene (SC308739). The cDNA was amplified and cloned into a pET-15b vector to attach the N-terminal His_6_-tag required for purification. The pET-15b construct containing *ptp4a3* sequence was confirmed by sequencing then transformed into *E. coli* BL21 (DE3) and purified on a TALON Metal Affinity Resin (Takara, Mountain View, CA, USA) column. Assays were performed using 6,8-difluoro-4-methylumbelliferyl phosphate (DiFMUP) as an artificial substrate at 25 °C for 30 min in 40 mM Tris-HCl (pH 7.0), 75 mM NaCl, 2 mM EDTA, and 4 mM DTT buffer. The reaction was carried out in 15 μL total volume per well of a black 384-well plate and initiated upon addition of DiFMUP at a final concentration of 12 μM (3× the K_m_ of PTP4A3 for DiFMUP, to ensure that the reaction velocity remained constant throughout the assay) to each well containing 1 μg of full-length recombinant human PTP4A3 protein. The fluorescence was measured using a SpectraMax M5 plate reader at 358 nm excitation and 455 nm emission. Fluorescence values were used to calculate the percent inhibition of enzyme activity relative to maximal activity, PTP4A3 in the absence of inhibitor, and maximal inhibition, PTP4A3 in the presence of 2 mM Na_3_VO_4_.

### 2.5. Quantitative RT-PCR

Total RNA was extracted from OvCa cell lines using the RNeasy Mini kit (Qiagen, Germantown, MD, USA) according to the manufacturer’s instructions. mRNA (1 μg) was treated with DNase to eliminate genomic DNA and then converted to cDNA using the RT^2^ first strand kit (Qiagen). Validated primer sequence pairs used for amplification of target genes were purchased from Qiagen and used with SYBR Green/ROX qPCR mastermix. Real-time monitoring of reactions was performed on BioRad CFX Connect Real-Time PCR detection system. PCR was performed by incubating at 95 °C for 10 min, followed by 39 cycles of 95 °C for 15 s, 60 °C for 1 min. Additionally, a melt curve analysis was included to ensure the absence of non-specific primer binding. Data were normalized to mRNA levels of hypoxanthine phosphoribosyltransferase 1 (HPRT) alone or the average of HPRT, actin, and GAPDH.

### 2.6. Western Blot Analysis

Cells and tissues were lysed in M-PER (Thermo Scientific, Waltham, MA, USA) containing 1x Halt protease inhibitor cocktail (Thermo Scientific) and quantified by Pierce BCA protein assay (Thermo Scientific). A total of 20 µg of total protein was separated using Novex NuPAGE SDS reagents (Invitrogen) and transferred to PVDF membranes. Membranes were blocked in 5% non-fat milk and 2% equine serum in TBS-t for 1 h, and then incubated with the primary PTP4A3 antibody in TBS-t for 1 h followed by secondary HRP-linked anti-mouse antibody in TBS-t for 1 h. Blots were stripped for 3 min using Restore PLUS stripping buffer, rinsed with TBS-t, blocked with 5% non-fat milk in TBS-t for 1 h, then incubated with loading control α-tubulin antibody in TBS-t for 1 h followed by secondary HRP-linked anti-antibody in TBS-t for 1 h. We used the following commercially available primary and secondary antibodies: PTP4A3 (Novus Biologicals, Centennial, CO, USA, catalog # MAB3219), anti-mouse IgG HRP-linked (Cell Signaling Technology, Danvers, MA, USA, catalog #7076) α-tubulin (Cell Signaling Technology, catalog #2125), anti-rabbit IgG HRP-linked (Cell Signaling Technology, catalog #7074). Bands were visualized using Immobilon Western Chemiluminescent HRP Substrate (Millipore, St. Louis, MO, USA) and the G-Box imaging system (Syngene, Frederick, MD, USA). Densitometry was calculated using Image Studio Lite software (Li-Cor, v5.2) (Lincoln, NE, USA).

### 2.7. Cytotoxicity Assay

We determined cytotoxicity using the CellTiter Glow 3D (Promega, Fitchburg, WI, USA). Cells were seeded (20 µL) in complete OvCa cell culture medium into 384 well ultralow attachment spheroid microplates (Corning, Corning, NY, USA): 250 cells/20 µL were used for A2780, COV362, COV362-47R, Kuramochi, OVCAR4, OVSAHO, V581, OVCAR8 cells and 500 cells/20 µL for OVCAR3 cells. Plates were incubated at 37 °C and 5% CO_2_ for 24 h to allow for spheroid formation. For single agent cytotoxicity studies, spheroids were exposed to 10 concentrations to determine the EC_50_ values. For the compound combination studies, we added the approximate EC_50_ concentration of JMS-053 (2.5 μL) to the plates 24 h after cell addition. Immediately after JMS-053 addition, vehicle or the second compound (2.5 μL) was added at the approximate EC_50_ concentrations. Each well contained 0.5% DMSO and 25 μL total volume after compound addition. Microtiter plates were incubated for 48 h (37 °C, 5% CO_2_) after compound addition. Subsequently, 25 μL of the CellTiterGlo 3D reagent was added to each well. Plates then incubated for 30 min with gentle shaking at room temperature. Luminescence data were captured on a TECAN Genios Pro (Baldwin, CA, USA) or a SpectraMax M5. For the single agent EC_50_ determinations, we used GraphPad version 9 (San Diego, CA, USA) and, for the compound interactions studies, we used the combination index values calculated using CalcuSyn version 2.11 software (Biosoft, Cambridge, UK).

### 2.8. In Vitro Scratch Wound Healing Assay

A2780 or OVCAR4 cells were seeded at 2.5 × 10^5^ cells per well in 24-well tissue culture plates and were cultured to confluence. Cells were then serum starved in RPMI medium supplemented with 2% FBS for 24–36 h. Each well was scratched longitudinally and horizontally with a pipette tip and then incubated for 14–18 h to allow gap closure. Cell migration images were captured with an EVOS XL Core and ImageJ Fiji software was use to determine wound closure by measuring the gap distance between cell fronts following inward migration versus the gap at the initial scratching.

### 2.9. In Vivo OvCa Dissemination Assay

Because PTP4A3 has been associated with cancer migration, adhesion and metastasis [21,22,23], we used a recently described model of OvCa dissemination [23] to assess the potential of JMS-053 to disrupt this critical cancer process in vivo. Exponentially growing taxane-resistant SKOV3-TRip2 cells were harvested and washed twice with 1xPBS. The cell suspension was adjusted to 2 × 10^7^ cells/mL in 2.1 mL of 1 × PBS to which an equal volume of growth factor reduced Matrigel was added. To the cell suspension, we added 1 μL DMSO vehicle or JMS-053 (final concentration of 5 μM) and incubated the cells for 1 h. Female nude mice (6–8 wk old; ~20 gm) were placed into weight-matched groups of 10 and injected with the SKOV3-TRip2 cells (1 × 10^6^) i.p. in 100 uL. Mice were treated twice once at 48 h and a second time at 96 h after tumor cell inoculation i.p. with JMS-053 (15 mg/kg/day) or vehicle (30% Capitsol, 40% PEG400, 30% PBS) (100 μL). The mice were sacrificed 14 days after tumor cell injection. All in vivo procedures were performed using a University of Virginia approved IACUC protocol (protocol code 4135, approved 8/10/2020). Animal care was administered in accordance to guidelines established by the American Association for Accreditation of Laboratory Animal Care.

### 2.10. Statistical Analyses

Cell-based and in vitro biochemical data were analyzed using GraphPad Prism 9 Software (San Diego, CA, USA). Statistical significance was determined as described in the text. Survival data was mined using KMplot from Nagy et al. [24], limited to the serous ovarian cancer dataset of 98 stage I/II and 1023 stage III/IV patient samples provided by Gyorffy et al. [25] from TCGA (https://www.kmplot.com/analysis/index.php?p=service&cancer=ovar) accessed 10 June 2021. Survival plots focused on overall survival (OS) splitting patients by the median values.

## 3. Results

### 3.1. PTP4A3 Expression in Human Patient OvCa

PTP4A3 phosphatase mRNA is significantly elevated in a variety of cancer lineages (Figure 1A). In the 1,457 cell lines found listed in the Cancer Cell Line Encyclopedia, leukemias and myelomas had the highest expression as measured by RNAseq among the 40 lineages examined but it is notable that the 55 OvCa cell lines were the thirteenth highest lineage in PTP4A3 expression with a mean mRNA expression level 3.8-fold higher than that seen in the human cancer line collection. qPCR profiling of established HGSOC cell lines, including Kuramochi, COV362 and OVCAR4, showed 5–20 fold higher PTP4A3 gene expression versus the nonmalignant control cell line HIO-180 (Figure 1B and Appendix A). Among the non-HGSOC cell lines both A2780 (i.e., endometrioid cancer) and A2780cp20 also had detectable PTP4A3 mRNA levels. PTP4A3 mRNA levels were higher in the drug resistant COV362-47R relative to the parental COV362 cells (Figure 1B) (*p* < 0.05) consistent with the suggestion that drug resistance is associated with elevated PTP4A3 expression [26], although this did not appear to be translated into higher protein levels (Figure 1C and Appendix A). HGSOC is considered to be intrinsically drug-resistant and of the HGSOC established cell lines, Kuramochi cells appeared to have the highest mRNA levels of all of the cells examined while OVCAR4 cells had the highest PTP4A3 protein levels (Figure 1C and Appendix A). Our OVCAR4 cells that were engineered to have depleted PTP4A3 by CRISPR/Cas9 were a pooled population and showed an ~50% decrease in PTP4A3 protein (Figure 1C and Appendix A).

A bioinformatic analysis using the KMplot Survival Database revealed that high PTP4A3 expression was associated with poor progression-free survival in 1,323 sampled epithelial OvCa (*p* < 0.0001, hazard ratio (HR) = 1.35; 95% Confidence Intervals = 1.17–1.56). In HGSOC early-stage disease, PTP4A3 expression levels were also correlated with overall survival (stage I/II- *p* = 0.0013, HR = 3.2; 95% Confidence Intervals = 1.22–8.34) (Figure 2A), with the 50 patients expressing low PTP4A3 surviving twice as long as the 48 high expressing patients (upper quartile survival in low expression cohort = 81.2 months compared to high expression cohort = 44.3 months) (Figure 2A). The beeswarm plot (Figure 2B) reflects the expression data obtained from the adjacent Kaplan-Meier plot. This survival difference was also seen in the 1001 patients with advanced disease (stage III/IV *p* = 0.0027, HR = 1.29; 95% Confidence Intervals = 1.09–1.52) (Figure 2C,D). In the TCGA dataset from tumors obtained from 300 patients with HGSOC, PTP4A3 was found highly expressed in 20% of the samples compared to 13% for MYC (Appendix A). In only 7 of the samples were both PTP4A3 and MYC mRNA levels elevated indicating no coordinated co-expression (Appendix A). As anticipated, there was no evidence of PTP4A3 mutations but amplification of the PTP4A3 gene was seen in 25% of the samples and we speculate this was likely responsible for the elevated mRNA levels (Appendix A). The top 20 genes that were co-expressed with PTP4A3 were also located on chromosome 8q24.3 and surprisingly MYC was not among them (Appendix A). Interestingly, most of the products of these genes have not been considered highly oncogenic. It is interesting that, unlike PTP4A3, overexpression of c-MYC in early stage HGSOC actually appeared to be associated with better overall survival further supporting the independence of these two gene products (Appendix A).

### 3.2. Chemical Structures of New Analogs and In Vitro Inhibition of PTP4A3

In an effort to further expand on the structure-activity relationship expansion of the thienopyridone core that we previously reported [17,27,28], we synthesized additional analogs of JMS-053 with side chain modifications to the thienopyridone core structure (Figure 3). Replacement of the imine with a carbonyl group did not markedly alter in vitro potency as an inhibitor of PTP4A3, i.e., IC_50_ values of EJR-912-41 (63.7 ± 7.3 nM, SEM) and EJR-912-57 (89.7 ± 14.0 nM) were similar to those of EJR-912-43 (53.7 ± 12.4 nM) or EJR-912-50 (60.6 ± 6.0 nM), respectively. In contrast, replacement of the imine with a bromide as in NRT-902-38 marked reduced potency in the PTP4A3 in vitro assay and this compound was not studied further. Placement of the pendant aryl moiety distal to the thiol group yielded only a slight reduction in phosphatase inhibition. These results suggested there was considerable tolerance to alter the location and structural nature of the core heterocycle substituents for further analog development.

### 3.3. In Vitro Cytotoxicity of JMS-053

We previously demonstrated that JMS-053 was cytotoxic with nine OvCa cell lines grown as pathologically relevant 3-dimensional spheroids but demonstrated no significant cytotoxicity against normal fibroblast (IMR-90) and human ovarian epithelial (HIO-180) cells with concentrations up to 25 µM [16]. To expand our exploration of HGSOC cells, we examined six human HGSOC cell lines, including a recently derived carboplatin resistant cell line, COV362-47R, and a patient-derived cell line V581 (Table 1 and Appendix A).

We incubated spheroids with compounds for 48 h to parallel our previous studies [16,29]. JMS-053 had low micromolar EC_50_ cytotoxicity values with all of the HGSOC cells tested and was more potent than any of the other analogs. JMS-053 also was more potent than the clinically used drugs olaparib and veliparib in all of the cell lines tested in vitro, although JMS-053 appeared to be less potent that topotecan in two cell lines. We only assayed paclitaxel at concentrations up to 5 μM and we were unable to establish an EC_50_ value. We previously identified NRT-870-59 as a potentially promising next generation analog based on its in vitro phosphatase inhibition [29], but it was less potent against the HGSOC cell lines, which lead us to deprioritized this compound. Because of the superior results with JMS-053 with HGSOC cell lines, we focused on this compound.

### 3.4. Cell Migration Assays

PTP4A3 has been previously been implicated in OvCa cell migration and invasion [16]. Thus, we further validated the role of PTP4A3 in human OvCa A2780 cell migration using JMS-053, which caused a robust concentration-dependent reduction in migration with an EC_50_ of 250 nM (Figure 4A). IL-6 has been reported to induce PTP4A3 expression and to promote OvCa tumor cell migration [30,31]. The IL-6-mediated PTP4A3 induction has been proposed to be mediated by STAT3 activation and SHP2 phosphatase repression [31]. As noted in Figure 4B, IL-6 markedly stimulated the migration of OVCAR4 cells. To further validate PTP4A3 in HGSOC cell migration, we use CRISPR/Cas9 methodology to deplete PTP4A3 in OVCAR 4 cells. The loss of PTP4A3 in the pooled population was ~50% than the parental cells (Figure 1C) but this was sufficient to reduce the basal migration of OVCAR4 cells and maredly reduce the IL-6 stimulated cell migration further validating the role of PTP4A3 in controlling the movement OvCa cells (Figure 4B). JMS-053 inhibited the migration of OVCAR4 wildtype cells in a concentration-dependent manner with an EC_50_ of ~500 nM (Figure 4C) but did not alter the migration of OVCAR4 PTP4A3 null cells (Figure 4D). These data demonstrate high PTP4A3 presence was required for the pharmacological actions of JMS-053.

### 3.5. Drug Combinations with JMS-053

We next used three established OvCa cell lines to examine formally the potential for synergistic interactions between JMS-053 and two drugs that are clinically used to treat OvCa, paclitaxel and olaparib, as well as a previously described inhibitor of the STAT3 pathway 864,669 [20] and a SHP2 inhibitor SHP009 [4], because of the reported role of STAT3 and SHP2 phosphatase in a feedforward loop in multiple myeloma [31]. In all of these studies, we used the approximate EC_50_ concentrations based on monotherapy exposure to determine the Combination Index using the method of Chou and Talalay [32]. We observed a marked synergy when JMS-053 was combined with paclitaxel in OVCAR8 cells, as indicated by the Combination Index of 0.59, and additivity in OVCAR3 and A2780 cells (Table 2). The poly-ADP-ribose polymerase inhibitor olaparib also showed additivity in the homologous competent A2780 cells with a Combination Index of 0.85 and exhibited some antagonism in the homologous recombination competent OVCAR3 and the homologous recombination deficient OVCAR8 cells [33] as indicated by the Combination Index of 1.69 and 2.10, respectively. The STAT3 inhibitor 864,669 when combined with JMS-053 was antagonistic in OVCAR3 cells and essentially additive in A2780 and OVCAR8 cells. The combination of JMS-053 with the SHP2 inhibitor SHP099 was antagonistic in A2780 cells.

### 3.6. Reduction of In Vivo Drug-Resistant OvCa Cell Dissemination by JMS-053

Because JMS-053 inhibited the movement of OvCa cells and exhibited synergistic and additive effects with paclitaxel, we next examined the ability of JMS-053 to block OvCa dissemination using SKOV3-Trip2 cells, because they represent a model of OvCa dissemination and they are paclitaxel resistant. Moreover, we previously found SKOV3-TRip2 cells to be two-fold more sensitive than the parental SKOV3 cells to JMS-053 in vitro [16]. We used a previously described OvCa model of tumor dissemination [23] but with SKOV3-TRip2 cells. Mice were examined 14 days after i.p. SKOV3-TRip2 cell implantation and some of the vehicle control mice did not exhibit detectable tumors. Nonetheless, even a relatively brief two-day treatment with JMS-053 resulted in a marked decrease in the number of implanted tumors found in the peritoneal cavity of the treated mice (Figure 5). These results further support a role of PTP4A3 in tumor dissemination and metastasis.

## 4. Discussion

Our current results complement previous reports [16,34,35] and demonstrate that high PTP4A3 expression is common in many cancer cell lines including OvCa, which in the USA is the most fatal gynecological cancer and the fifth most common cancer-related cause of death in women [30]. Unlike many other oncogenic proteins, PTP4A3 is rarely mutated; rather, the high protein levels result from gene amplification, increased transcription and translation, and possibly altered protein degradation [11]. It is noteworthy that PTP4A3 has been reported to be induced by genotoxic stress caused by cancer chemotherapeutic agents such as cisplatin, etoposide and doxorubicin [26]. We found that cell lines derived from HGSOC tumors express higher PTP4A3 mRNA levels than other histologic subtypes of OvCa or nonmalignant HIO-180 cells and had high PTP4A3 protein levels (Figure 1C). Kuramochi cells, which have previously been identified as highly representative of human HGSOC [36], had among the highest mRNA levels. TP53 mutations are seen in a majority of the HGSOC cells in the Cancer Cell Library Encyclopedia [36] and its gene product, p53, regulates PTP4A3 expression [37]. Previous genomic studies have revealed three major common genetic features of HGSOC [36]. First, HGSOCs have extensive copy number alterations, which includes the region encoding *ptp4a3*, namely chr8q24 [7,36]. This gene amplification is likely responsible for the high PTP4A3 levels found in many HGSOC cells. Kuramochi cells, for example, have a copy number variation score from COSMIC of 2. Second, TP53 mutations are almost universal, which would make a compound such as JMS-053 that appears to act on cells with mutant or deleted TP53, potentially desirable. Third, somatic mutations in protein coding regions are uncommon with the exception of BRCA1, BRCA2 and TP53, making drugs that function on both homologous recombination-deficient and -proficient cells desirable. Kuramochi, COV362 and OVSAHO are all BRCA mutated while OVCAR4 is not, and JMS-053 efficacy showed no significant difference among them when examined collectively. Kuramochi and OVSAHO were the least sensitive to JMS-053 treatment and this could reflect amplification levels of 8q24, as both lines contain focal amplification around c-MYC. Most HGSOC tumors are TP53 mutant and JMS-053 sensitivity varied independent of TP53 status. A2780ip2 and A2780cp20 are a parental and resistant cell line pair where the resistance to platinum was accompanied by a mutation of TP53. There were differences between the resistant and parental lines but when compared to other TP53 mutant cells the association was not significant.

The lack of common oncogenic drivers and the relative late detection of HGSOC in women create a therapeutically challenging disease. The highly oncogenic PTP4A3 coordinates many pathways and processes associated with poor patient prognosis in all stages of HGSOC, including migration, invasion, survival, and chemoresistance. OvCa cells are known to continuously secrete cytokines that promote tumorigenicity in both autocrine and paracrine fashions while also receiving signals from the tumor microenvironment [30]. IL-6 activates STAT3, which has previously been shown to enhance tumor cell growth, resistance to chemotherapy, and tumor dissemination in some tumor types [31,38]. In the current study, we demonstrate genetically and pharmacologically for the first time that PTP4A3 enables HGSOC cell migration (Figure 4). CRISPR/Cas9 mediated loss of PTP4A3 in HGSOC cells markedly decreased IL-6 mediated cell migration. The selective and allosteric PTP4A3 inhibitor JMS-053 reduced IL-6 mediated migration only in cells expressing PTP4A3, indicating a requirement for the putative intracellular target. JMS-053 was also capable of reducing drug-resistant OvCa peritoneal dissemination in a mouse model.

JMS-053 showed considerable efficacy in multiple HGSOC lines, including cell lines resistant to current standard of care chemotherapeutics and a patient-derived HGSOC cell line, V581, and was more potent than olaparib and veliparib in our acute 3-dimentional cytotoxicity assay. These observations complement our previous cellular and in vivo studies with JMS-053 and OvCa cell lines that were not HGSOC [16]. Chemical modifications of JMS-053 established new insights on structure-activity relationships but did not result in analogs with greater cellular activity or identify alternative candidates for monotherapy in HGSOC. We did, however, observe evidence of synergy with paclitaxel and JMS-053 in OVCAR8 cells (Table 2), which warrants further investigation.

In summary, our results support a role for PTP4A3 in the progression of human OvCa and provide further evidence that inhibitors of the oncogenic phosphatase should be pursued for clinical development. It will be important to identify potential biomarkers that will allow us to select individuals with OvCa who would benefit most from treatment with a PTP4A3 inhibitor.

## Figures and Tables

**Figure 1 biomolecules-11-00969-f001:**
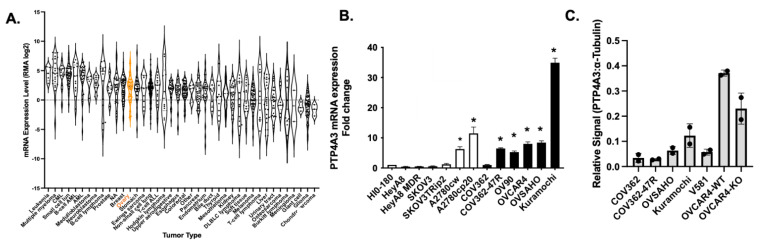
PTP4A3 is overexpressed in OvCa cell lines. (**A**) PTP4A3 mRNA expression in the Broad Cancer Cell Library Encyclopedia. Data obtained from https://portals.broadinstitute.org/ccle/ accessed on 31 June 2021. Ovarian cancers are indicated in orange. (**B**) PTP4A3 mRNA levels measured by qPCR and normalized to HPRT were elevated in all HGSOC cells lines (black columns) compared to HIO-180 nonmalignant epithelial cells grown in culture with the exception of COV362 cells. N = 3, bars = SEM. * *p* < 0.05 ANOVA. Appendix A demonstrates the differences are not associated with the HPRT reference gene product. (**C**) Detection of PTP4A3 protein in high grade serous OvCa cell lines. Fifty µg of total protein lysate was loaded per lane and protein levels were detected by Western blots with PTP4A3 antibody. Mean values of 2 technical replicates. Bars = range.

**Figure 2 biomolecules-11-00969-f002:**
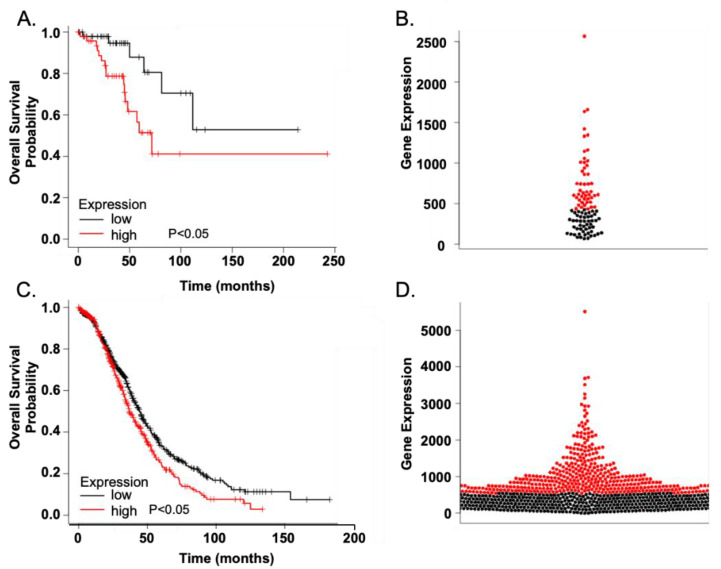
PTP4A3 overexpression in human OvCa is associated with poor survival, even in early stage disease. (**A**) High PTP4A3 expression was associated with poor survival in tumors from 98 patients with stage I/II serous OvCa. Red symbols and line represents the 48 tumor samples with high PTP4A3 expression and black symbols and line represent the 50 tumor samples with low PTP4A3 expression. Patients with low PTP4A3 gene expression showed almost double the length of overall survival: 81.2 vs. 44.3 months. *p* < 0.05. (**B**) Beeswarm plot showing high expression of PTP4A3 even in early stage OvCa patients. Red and black symbols are from Panel A. (**C**) High PTP4A3 expression is also associated with reduced survival in stage III/IV serous OvCa patents. Red symbols and line represents the 394 tumor samples with high PTP4A3 expression and black symbols and line represent the 629 tumor samples with low PTP4A3 expression. Patients with low PTP4A3 expression had an overall survival of 44.5 months compared to 36.8 months for patients with high PTP4A3 expressing tumors. *p* < 0.05. (**D**) Beeswarm plot showing high expression of PTP4A3 maintained in a significant population of patients in later stage disease. Red and black symbols are from Panel (**C**).

**Figure 3 biomolecules-11-00969-f003:**
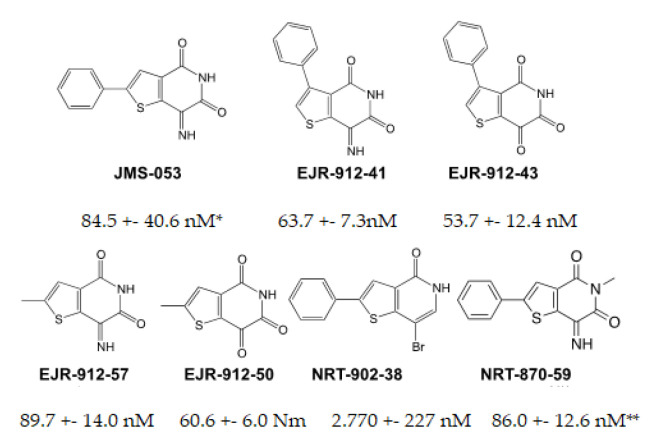
Chemical structures of the inhibitors and in vitro IC_50_ against recombinant human PTP4A3. Mean ± SEM, N = 3, unless indicated by * N = 6 or ** N = 8.

**Figure 4 biomolecules-11-00969-f004:**
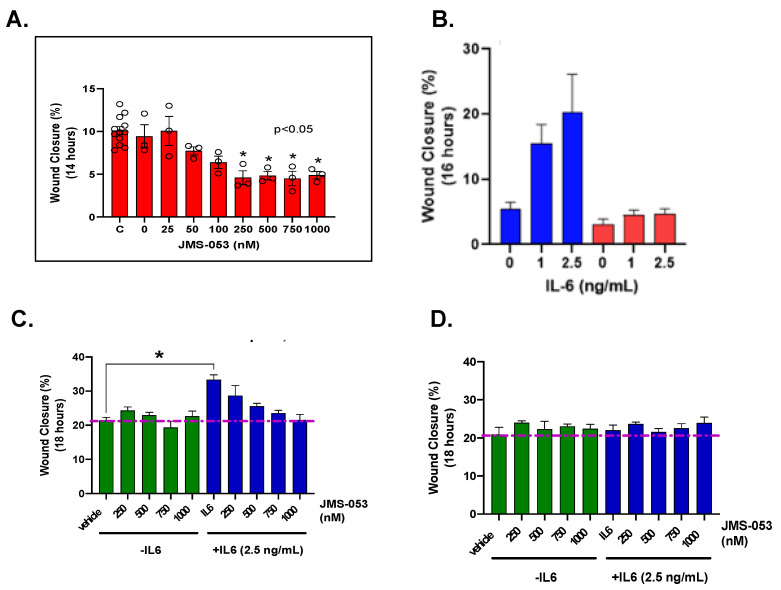
Inhibition of PTP4A3 activity blocks OvCa cell migration. (**A**) JMS-053 inhibits migration of A2780 cells. A2780 were cultured to confluence then “wounded” and treated with vehicle control or JMS-053. Wound closure was quantified at 14 h post-wounding. Mean ± SEM (**B**) IL-6 caused a concentration-dependent migration in PTP4A3 wildtype OVCAR4 cells (blue). CRISPR/Cas9 knock-out of PTP4A3 in OVCAR4 cells reduces IL-6 mediated cell migration (red). Mean ± SEM. (**C**) JMS-053 causes a concentration-dependent inhibition of IL-6-mediated migration in wildtype OVCAR4 cells. Wound closure was quantified at 18 h post-wounding. Mean ± SEM. N = 3. Vehicle control indicated by the red line. (**D**). With OVCAR4-PTP4A3 knockout cells, JMS-053 did not cause any further inhibition of migration in the presence of IL-6. Wound closure was quantified at 18 h post-wounding. Mean ± SEM, N = 3. * *p* < 0.05.

**Figure 5 biomolecules-11-00969-f005:**
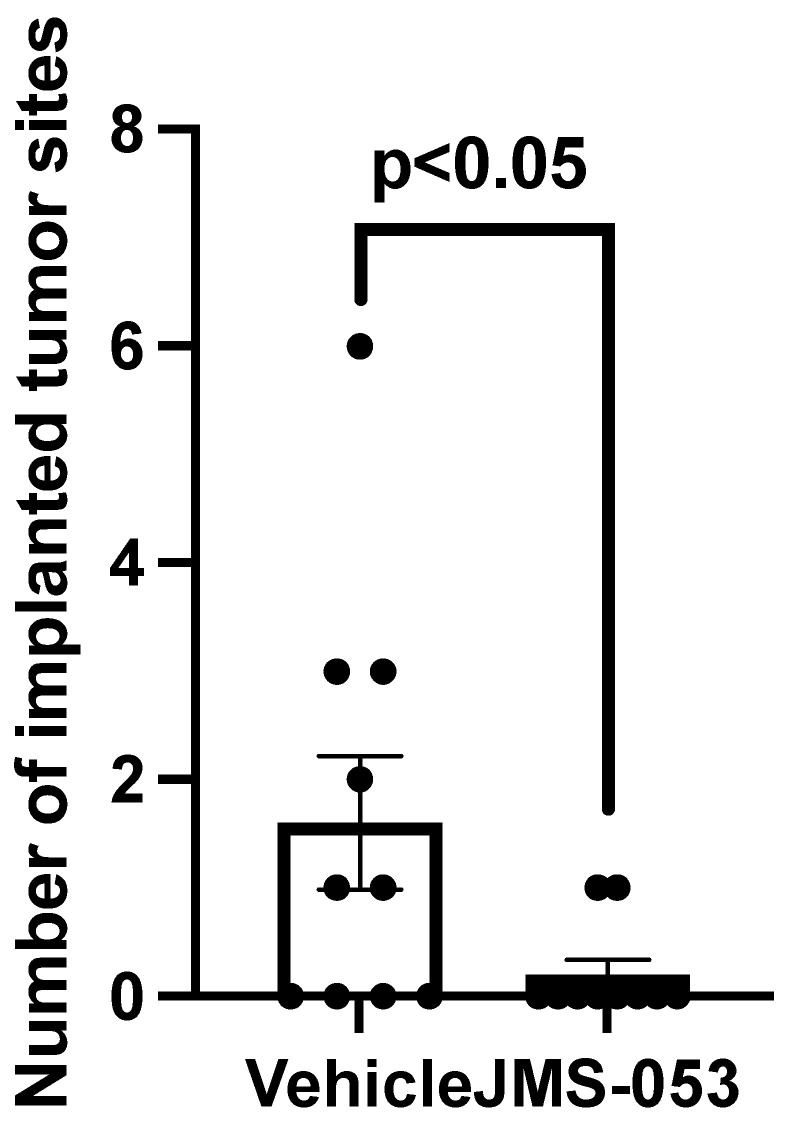
Reduction in i.p. tumor dissemination by JMS-053. Mice were injected i.p. with paclitaxel-resistant SKOV3-TRip2 cells that were briefly (1 h) incubated with 5 μM JMS-053 or vehicle and followed with i.p. treatment with 15 mg/kg JMS-053 one and two days later. Mice were sacrificed 14 days after tumor cell injection and the number of i.p. tumors counted. Individual values are indicated as well as the mean value ± SEM. Unpaired Student *t*-test.

**Table 1 biomolecules-11-00969-t001:** Cytotoxicity of compounds against HGSOC cell lines grown in 3D as spheroids. N = 3 biological replicates with 3 technical replicates in each assay. ^a^ N = 2 biological replicates ± range with 3 technical replicates in each assay. ^b^ N = 1 with 3 technical replicates. ^c^ ND = not determined.

Compound	EC_50_ (μM ± SEM)					
	COV362	COV362-47R	Kuramochi	OVCAR4	OVSAHO	V581 ^c^
JMS-053	14.2 ± 0.5	24.9 ± 0.1 ^a^	6.1 ^b^	4.8 ± 2.0	13.9 ± 3.5	7.4 ^b^
NRT-870-59	>50	>50	30.6 ^b^	>50	>50	>50
EJR-912-41	31.6 ± 6.8	>50	7.0 ^b^	26.9 ± 3.9	38.7 ± 1.7	>50
EJR-912-43	>50	>50	>50	>50	>50	>50
EJR-912-50	>50	>50	>50	>50	>50	>50
EJR-912-57	28.3 ± 3.1	>50	>50	>25	>50	>50
Paclitaxel	>5	>5	>5	>5	>5	>5
Olaparib	>50	>50	>50	>50	>50	>50
Veliparib	>50	>50	>50	>50	>50	>50
Topotecan	2.5 ± 0.7	25.3 ± 9.8	ND	2.5 ± 0.8	2.5 ± 1.1	ND

**Table 2 biomolecules-11-00969-t002:** Compound combinations in human OvCa cells. Combination Index (CI) was measured with EC_50_ concentrations of each compound. The single agent EC_50_ values for A2780, OVCAR3 and OVCAR8 cells in these studies were: JMS-053: 3, 5 and, 10 μM; paclitaxel: 0.04, 60 and 10 μM; olaparib: 50, 90, 70 μM; 864,669: 7, 70, 30 μM; SHP099: 0.5, 60, 30 μM. N = 3. Unpaired Student’s *t*-test compared to JMS-053 alone. * *p* < 0.05.

Compound	Combination Index ± SEM
A2780	
Paclitaxel	0.90 ± 0.18
Olaparib	0.85 ± 0.13
864,669	0.87 ± 0.08
SHP099	2.55 ± 1.10 *
OVCAR3	
Paclitaxel	1.04 ± 0.18
Olaparib	1.82 ± 0.14 *
864,669	1.69 ± 0.12 *
SHP099	1.28 ± 0.04
OVCAR8	
Paclitaxel	0.59 ± 0.07 *
Olaparib	2.10 ± 0.12 *
864,669	1.46 ± 0.09
SHP099	1.49 ± 0.06

## Data Availability

All primary data are available to qualified individuals by contacting the contect authors.

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
