# Peer review of "Credentialing and Pharmacologically Targeting PTP4A3 Phosphatase as a Molecular Target for Ovarian Cancer"

_biomolecules, 2021, doi:10.3390/biom11070969_

Round 1

Reviewer 1 Report

This report describes a preclinical study describing the potential role of the phosphatase PTP4A3 in ovarian cancer and testing a first-generation phosphatase inhibitor in ovarian cancer cell-based and xenograft models. The manuscript is generally well-written.

            Specific comments:

  1. Lines 45-48: This statement is largely incorrect. Phosphatases have not been ignored as drug targets. Phosphatases were the next in line as targets after the community became saturated with kinase inhibitors; however, phosphatases are difficult targets due to protein structure without obvious small molecule binding sites and important roles of phosphatases in normal tissue as well as tumors. The sentence should be modified.
  2. Lines 85-88: The vendor for OVCAR4 cell need only be mentioned once.
  3. Lines 183-184: 48 hours is a very short incubation time especially for a 3D cell-based model system. Justify the very short exposure time. Are the compounds unstable in cell culture medium?
  4. Lines 178-189: Why was EC50 used rather than IC50? Are no concentration response curves shown because full concentration response data were not generated? The cell-based data in Table 1 look unusual. Be clear that the drug exposure time was very short, perhaps too short for some drugs to manifest their full effect. Justify the use of EC50s.
  5. Lines 200-209: Implantation of ovarian cancer xenograft cells ip to grow peritoneal nodules is not uncommon. It appears that mice were given a single injection of the compound or of the vehicle, state clearly that the in vivo effect was due to a single injection of the compound.
  6. Table 2: The table legend and the table contents are not clear. Why are the various drug concentrations presented oddly out of order and over such a limited range? It would be very helpful to show concentration response data for the single agents and the combinations.
  7. Figure 5: It appears that the tumor take rate in the controls was very poor or that the vehicle used was toxic when administered ip. The mean difference between the vehicle and treated groups appears to be less than 2 versus less than 1 nodule. If the vehicle was toxic to the tumor cells, these data should be removed or at least, an untreated control group should be added. Those data are likely available and were generated during development of the model.

Author Response

Reviewer #1. We thank the Reviewer for his or her complimentary comments concerning our manuscript including that it was “well-written”.

  1. Lines 45-48: This statement is largely incorrect. Phosphatases have not been ignored as drug targets. Phosphatases were the next in line as targets after the community became saturated with kinase inhibitors; however, phosphatases are difficult targets due to protein structure without obvious small molecule binding sites and important roles of phosphatases in normal tissue as well as tumors. The sentence should be modified.
    Response: As recommended, we have modified (new) Lines 46-51 to now read: “the PTPs have been more challenging therapeutic targets, in part, because their protein structures often lack obvious small molecule binding sites and they have important roles in normal tissues as well as tumors [2,3]. Recently, however, allosteric small molecule inhibitors of two PTPs, PTP1B and SHP2 phosphatase, have made significant advancements in clinical trials and this has generated considerable enthusiasm for PTPs as relevant and innovative molecular targets for antineoplastic interventions [4-6].”

  2. Lines 85-88: The vendor for OVCAR4 cell need only be mentioned once.
    Response: This has been corrected.

  3. Lines 183-184: 48 hours is a very short incubation time especially for a 3D cell-based model system. Justify the very short exposure time. Are the compounds unstable in cell culture medium?
    Response: We used a 48 hour incubation time because that was the exposure time used in our previous publications (Reference 16, McQueeney et al., Oncotarget 2018, 9:8223-8240 and Reference 29, Lazo et al., J Pharmacol Exp Ther 2019, 371:652-662) and we wished to harmonize our previously published results with the current results. Moreover, we were interested in the acute cytotoxic effects of the compounds and wished to align the cytotoxicity studies with the incubation times for the migration studies shown in Figure 4. We agree longer incubation times with the compounds might result in even lower cytoxicity EC50 values but we were mindful of issues associated with longer incubations with larger sized spheriod, including reduced compound penetration and the hypoxia, nutrient deprevation and increased acidity seen with larger spheriods. We have now added new text to justify the use of a 48 hour incubation time (Lines 315-316). Nonetheless, the Reviewer’s thoughtful comments have inspired us to conduct new experiments with longer incubation times, which might demonstrate that our compounds are even more potent than what we have observed with the acute assay. These experiements, however, will require several months to complete and will be part of a future manuscript. We have examined the stability of JMS-053 in cell culture medium and it is stable for at least 3 days.
  4. Lines 178-189: Why was EC50 used rather than IC50? Are no concentration response curves shown because full concentration response data were not generated? The cell-based data in Table 1 look unusual. Be clear that the drug exposure time was very short, perhaps too short for some drugs to manifest their full effect. Justify the use of EC50s.
    Response: We thank the Reviewer for this thoughtful comment. We have used EC50 rather than IC50because we were measuring cell death in the 3D spheroid assay not just inhibition of cell proliferation (mentioned on Lines 95-96). This nomenclature is consistent with our previous usage in References 16 and 29. Table 1 reflects the results of multiple 10-point concentration-response curves. We noticed some errors in the previous version of Table 1, such as the EC50 values for topotecan, which may have been the source of some of the confusion and these have been corrected in the new Table. In addition, in response to the Reviewer’s query, we have now inserted Supplemental Figure 5 showing representative concentration-response curves for compounds with OVCAR4 cells. We elected not to include concentration-response curves for the other cell lines because they are mostly negative (i.e., flat) and would add a total of 60 additional panels (6 cell lines x 10 drugs or compounds) to the Supplemental Material. We do not believe this information would add to the manuscript.

  5. Lines 200-209: Implantation of ovarian cancer xenograft cells ip to grow peritoneal nodules is not uncommon. It appears that mice were given a single injection of the compound or of the vehicle, state clearly that the in vivo effect was due to a single injection of the compound.
    Response: We are sorry about this confusion and have modified the text to more clearly indicate that mice were treated twice, once at 48 and a second time at 96 h after implantation (Lines 208-210 and 386).

  6. Table 2: The table legend and the table contents are not clear. Why are the various drug concentrations presented oddly out of order and over such a limited range? It would be very helpful to show concentration response data for the single agents and the combinations.
    Response: We have modified the Table 2 legend to clarify the experimental design. Each EC50 value for the single agent was the result of a 10-point concentration curve and the CI value indicates in a concise manner the analyses of these results using the CalcuSyn software.
  7. Figure 5: It appears that the tumor take rate in the controls was very poor or that the vehicle used was toxic when administered ip. The mean difference between the vehicle and treated groups appears to be less than 2 versus less than 1 nodule. If the vehicle was toxic to the tumor cells, these data should be removed or at least, an untreated control group should be added. Those data are likely available and were generated during development of the model.
    Response: We thank the Reviewer for this observation and we agree that the tumor take was low in the vehicle treated mice but we believe this is likely an assay that closely simulates the early pathobiology of ovarian cancer dissemination. We do not have data on tumor dissemination with the SKOV3-TRip2 cells in these mice without vehicle treatment nor have we been able to identify any information in the literature on this topic. The formulation we used is an FDA-approved vehicle for use in humans so we think that is unlikely it is toxic to cells and we have seen no untoward effects when mice have been treated for two weeks i.p. with this volume of vehicle. We believe the low tumor take in the vehicle-treated mice more likely reflects the short period of time after i.p. tumor implantation when the mice were sacrificed, i.e., 14 days. We do not believe that it is appropriate to remove the data from untreated control group and would prefer that the reader view the entire data and make their own decision about the experimental results. The differences in tumor implantation between the two groups are statistically significantly different and we believe the data support our conclusion. We have now added new text mentioning the low tumor take in the vehicle treated mice (Line 581). Based on these promising in vivo data, however, we are now preparing to conduct new studies that will have a longer duration from i.p. implantation to allow micrometastases to be detected and we will use multiple compound treatment schedules as well as a non-vehicle control group in the future.

Reviewer 2 Report

The authors tried to understand better the role of PTP4A3, a tyrosine phosphatase in cancer.

More specifically, they found a correlation between the mRNA of PTP4A3 in ovarian cancer cell lines and the severity of the disease, which is manifested by the mortality of the women.

The synthesized new compounds to inhibit PTP4A3 and suggested these drugs might be beneficial for the treatment of ovarian cancer.

Questions: Fig 1 A shows a huge scatter of data. What could be the reason? Can valid conclusion be drawn? In Fig 1 A only mRNA is correlated. I could easily claim that protein is relevant for the activity of a phosphatase. Wouldn’t it be better to create on a protein basis or enzyme activity basis?

I see not statistical significances in Fig 2. Why?

In table 1: why is the cytotoxicity given with an n=1 or n=2? Is it possible to conclude anything in a meaningful way?

No statistics in Table 2. Why?

Fig. 5: it would be better to use p<0.05 throughout the manuscript.

Please put an asterisk in supplementary figure 1 where you expect PTP4A3,. What are the other bands on the lanes? Did you try blocking experiments?

What is the message of suppl. Fig 2? The data are impossible to read. Supple figure 4: how exactly did you normalize and why?

Author Response

Reviewer #2. We thank the Reviewer for his or her complimentary comments concerning our manuscript as well as the thoughtful questions.

  1. Questions: Fig 1 A shows a huge scatter of data. What could be the reason? Can valid conclusion be drawn? In Fig 1 A only mRNA is correlated. I could easily claim that protein is relevant for the activity of a phosphatase. Wouldn’t it be better to create on a protein basis or enzyme activity basis?
    Response: The results in Figure 1A are from the 1,457 cell lines found in the Cancer Cell Line Encyclopedia and the scatter likely reflects the heterogeneity inherently seen with the cultured cancer cell lines. The results are similar to what is seen in other smaller public databases, such as the NCI-60 cell lines. PTP4A3 mRNA levels are regulated by copy number, transcriptional activity and mRNA degradation. We fully agree that protein levels or enzymatic activity profiles would be informative, which is why we included Figure 1C. Unfortunately, PTP4A3 protein levels were not annotated in the Cancer Cell Line Encyclopedia cell lines and we are unaware of data on PTP4A3 levels in any large panel of human ovarian cancer cell lines. We also are unaware of data on PTP4A3 phosphatase activity in any large panel of human ovarian cancer cell lines.

  2. I see not statistical significances in Fig 2. Why?
    Response: We indicated the statistical analyses in the text (Lines 253-261) but have now also indicate the p values in the figure itself and the figure legend to emphasize the differences.

  3. In table 1: why is the cytotoxicity given with an n=1 or n=2? Is it possible to conclude anything in a meaningful way?
    Response: Our goal in Table 1 was to provide a comprehensive view of our compounds’ cellular activity with as many HGSOC cells as possible. The Kuramochi and V581 cells grow slowly and we could not obtain multiple biological replicates quickly. We do think the results are meaningful as each n (or biological replicate) represents the mean results of three technical replicates, which we have now indicated in the Figure legend. Therefore, we believe the information is useful when place alongside the other data. We could remove the Kuramochi and V581 data from the Table, if the Reviewer or Editor would like us to do, so but we think that its removal will reduce the impact of the Table.
  4. No statistics in Table 2. Why?
    Response: We have now added the requested statistical analyses.

  5. 5: it would be better to use p<0.05 throughout the manuscript.
    Response: We have made this suggested change.

  6. Please put an asterisk in supplementary figure 1 where you expect PTP4A3,. What are the other bands on the lanes? Did you try blocking experiments?
    Response: We have added the asterisk in the new Supplementary Figure 2. We do not know what the other bands are, although there is evidence in the literature that PTP4A3 can form higher order structures, namely trimers, which may be responsible for the ~70kDa bands. We have not tried blocking experiments but as we mentioned we now have PTP4A3 CRISPR/Cas9 OVCAR4 cells, which have lower levels of this ~20 kDa PTP4A3 band.

  7. What is the message of suppl. Fig 2? The data are impossible to read. Supple figure 4: how exactly did you normalize and why?
    Response: We have improved the visual presentation of the information in (now) Supplemental Figure 3, which indicates there is no co-overexpression of the known oncogene c-Myc with PTP4A3 and the genes that are mutually overexpressed. This is important for credentialing PTP4A3, because both PTP4A3 and c-Myc are located on chromosome q24.3. We think the information in the Supplemental Figure 3 (previously Supplemental Figure 2) is useful because it emphasizes the potential unique oncogenic role of PTP4A3 in HGSOC. We included Supplemental Figure 1 (previously Supplemental Figure 4) to exclude the possibility that the differences we observed in Figure 1B was not due to the denominator used in Figure 1B, namely HPRT. We averaged the mean individual values of the three so called “housekeeping” genes as the denominator to demonstrate the robustness of the data and independence from a single “housekeeping” gene. We added some additional text on Line 153 and in the legend of Supplemental Figure 1 to explain this in the revised manuscript.